# Source Code Changes Just-In-Time Update Via Code Semantics

*Abstract*—To tackle the trouble of incomplete, insufficient, or misaligned code comments during software development and maintenance, various techniques are emerged to modify the comments of plain language in accordance with code alterations. However, these methods have two significant limitations: addressing the source code involving non-temporal and long-rang dependencies poses challenges. With the aim of surpassing these restrictions, we present a novel approach named Code Comment Update (CCU) model, which incorporates self-attention, positional encoding, and relative positional representation to effectively capture the relationships between different source code tags. This allows it to effectively grasp extended and non-temporal interdependencies within the source code. The comment-update module of CCU produces fresh comments by harnessing the power of existing code alterations and comments. The results of Experiment demonstrate that CCU outperforms the three baseline methods in terms of metrics such as exact match, METEOR, BLEU, and SARI.

*Index Terms*—Code Changes, Code Comments, Code Semantic Learning, Code Comment Update

## I. INTRODUCTION

During software development, code comments exert a crucial influence [1]. They act as conduits for developers to document critical details, encompassing the objective, implementation, and utilization of code segments, alongside the connections and progression of the codebase. These code comments aid in enhancing program comprehensibility and fostering collaboration among developers, underscoring the significance of attaining a thorough comprehension of the codebase. However, despite their significance, code comments are often neglected by developers, and the introduced inconsistencies in code comments [2]. As a result, many comments become inconsistent or outdated, rendering them less useful. Inconsistencies between comments and code hinder maintenance efforts and can lead to future errors. Poor comments can raise development and maintenance expenses, and can represent adverse effect on system fault-tolerance. Therefore, promptly addressing inadequate comments or avoiding their introduction altogether is crucial.

Recent advancements in neural machine translation have spurred research in automatically producing fresh comments that are activated by code modifications [1]. This approach presents several benefits, such as the reduction or prevention of poor comments. The manual creation of code comments is a time-consuming and labor-intensive task, and ensuring comment quality can be challenging due to the informal nature of natural language when compared to source code. As a result, there is an urgent need for effective methods that automate the process of updating code comments.

Prior studies have explored various approaches to address the issue of inconsistent comments. Rule-based methods [2] have been employed to detect inconsistencies in specific cases. Other approaches, such as tcomment [3], focus on dynamically detecting inconsistencies but are limited to specific types of attribute detection. Comment generation methods [4] have the capability to produce comments that are derived from code, but may disregard existing relevant information. Comment UPdater (CUP) [5], a significant advancement in comment updating, utilizes NMT and a vast collection of code. In the realm of code comment updates, Tufano et al. [6] introduced a RNN based Seq2seq model. Their model aimed to learn code changes and provide assistance to developers by predicting potential code modifications that reviewers might suggest. However, two main hurdles endure in tackling the issue of code changes when it comes to updating code comments.

**Challenge of capturing long-range dependencies.** The challenge is posed by the effective capturing of the intricate relationships between distant elements in the source code. The vast size and intricate structure of the code make it difficult for existing RNN-based sequence-to-sequence models [6] to fully grasp the long-term dependencies within the code. The sequential processing of source code tokens hampers the model's ability to establish meaningful connections across distant elements, resulting in an incomplete comprehension of the source code. Moreover, as the length of the source code sequence grows, the model struggles to harness the entirety of the available information and fails to adequately incorporate the structural attributes of complex data. Consequently, the learning performance of the model is negatively affected.

**Non-timing dependence challenge.** During the process of code comment updates, the modifications made to code segments are often closely linked to their functionality. However, existing models encounter difficulties in effectively learning and incorporating these changes due to non-temporal dependencies arising from non-sequential interactions between words. Traditional timing-related models, such as RNN, lack the capability to capture these non-temporal dependencies, leading to inaccuracies in code comment updates. To overcome this limitation, it is crucial to develop innovative approaches that can effectively capture and leverage the non-temporal dependencies in code changes. This would enable more accurate and dependable updates to code comments.

To address the aforementioned challenges, we introduce a model named CCU (Code Comment Update), which comprises comment revision and code semantic understanding, to overcome the challenges of long-term and non-temporal

dependencies in code comment updates. We conducted our model training and evaluation using the same dataset as Panthaplackel et al. [1], which was constructed from an open-source Java project on GitHub. We conducted a comparative analysis of our model against three distinct categories of baselines, and employed several automated metrics that assess language generation tasks and tasks related to editing natural language text.

In summary, the contributions of this work are as follows.

(1) We proposed CCU to solve non-time-based and persistent correlations issues. For editing code with long sequences, strong structures, and close contextual relationships, the self-attention mechanism and relative position representation are used to modeling the correlation between source code tags in a pairwise manner, which overcomes the constraints of on-sequential correlation and persistent dependency in the source code.

(2) We conducted training and evaluation of the model using the dataset provided by Panthaplackel et al. [1]. The experimental results demonstrate that CCU surpasses the performance of the three baseline methods across four evaluation metrics: BLEU, exact match, SARI, and METEOR. Among them, the SARI, METEOR, exact match, and BLEU are respectively increased by 4.9%, 3.195%, 4.67%, 4.9%, and 0.15%, compared to the baselines. The results proves that our model is effective and practical.

## II. THE MODEL

Firstly, we provide a synopsis of CCU framework in Section II-A. Following that, we proceed to discuss the specifics of the components encompassed within CCU in Sections II-B and II-C.

### A. Overview

In the context of updating code comments, the objective is to modify a given comment in accordance with the changes made to a method. This involves providing the method, its associated comment, and the updated version of the method. To ensure that the updated comment accurately reflects the modifications in the code, we introduce CCU, which accounts for both the code modifications and the pre-existing comments. By simultaneously learning the representations of code changes and old comments, the CCU model effectively captures the relationship between them. The CCU model produces a sequence of revisions that can be applied to the initial comments, effectively reflecting the modifications made in the source code. Essentially, the model undergoes training to generate editing actions that steer the creation of a fresh comment by considering the changes made to the original code.

In Figure 1, A detailed overview of the CCU model for code comment updating is presented. The model comprises two main components: comment updating and code meaning learning. The code meaning learning component comprises of a decoder and two encoders. To encode the current comments, a bidirectional GRU is employed, considering the typically short length of comments. The contextual information of the

comments is captured by this encoder. For code changes, which often involve longer sequences and inter dependencies, a transformer encoder is utilized. The relationships between the code changes are effectively modeled by the transformer encoder, allowing long-term dependencies to be captured by the model. A series of editing actions based on the encoded information from the encoders is generated by the decoder, implemented as a GRU. These editing actions serve as guidance for updating the comment. The initial state of the decoder is formed by combining the concluding states of the two encoders, creating a comprehensive representation that encapsulates the code's information modifications and prior comments. In summary, the semantic relationship between code changes and existing comments is effectively learned by the CCU model, enabling accurate and informative updates to the comments based on the modifications in the source code. Subsequently, the GRU is employed to generate and train a sequence of comment modification operations, facilitating the execution of the comment editing process.The process of updating comments involves subsequently reranking them and parsing the editing sequence. The comment revision component examines and reorganizes the output of comment update operations generated by the code meaning learning component, resulting in the generation of refined and polished comments.While preserving the content that should remain unchanged,the existing comments' style attributes are retained, and new comments are generated accordingly.

---

**Algorithm 1** CCU For Comment Update Algorithm
___
**Input:** code edit $t$, old comment $x$
**Output:** new comment $X'$
  1: **for** each $t$ and $x$ **do**
  2:     Let $h_1 \leftarrow t$, Transformer as an encoder, input $t$ and output context vector $h_1$;
  3:     Let $h_2 \leftarrow x$, bidirectional-GRU as an encoder, input $x$ and output context vector $h_2$;
  4:     Let $H' \leftarrow h_1$, $h_2$. $h_1$ and $h_2$ are connected to form $H'$;
  5:     Let $T' \leftarrow H'$, GRU as a decoder, input $H'$ and output comment edit $T'$;
  6: **end for**
  7: Update comment based on old comment $x$ and comment edit $T'$;
  8: **for** each $T'$ **do**
  9:     Let $X' \leftarrow T'$, parse edit sequence and reranker to get $X'$;
 10: **end for**
 11: **return** $X'$

---

Algorithm 1 describes this process of ode comment updating in detail. The input is the code edit $t$ and the old comment $x$, and the output is a new comment $X'$ after the update. The first step is to train the model to generate comment edits. Through the encoder-decoder structure, source code changes and their relationships to existing comments are learned, so as to train a model to generate comment update operations related to code changes, thus the output is a comment update-related editing of

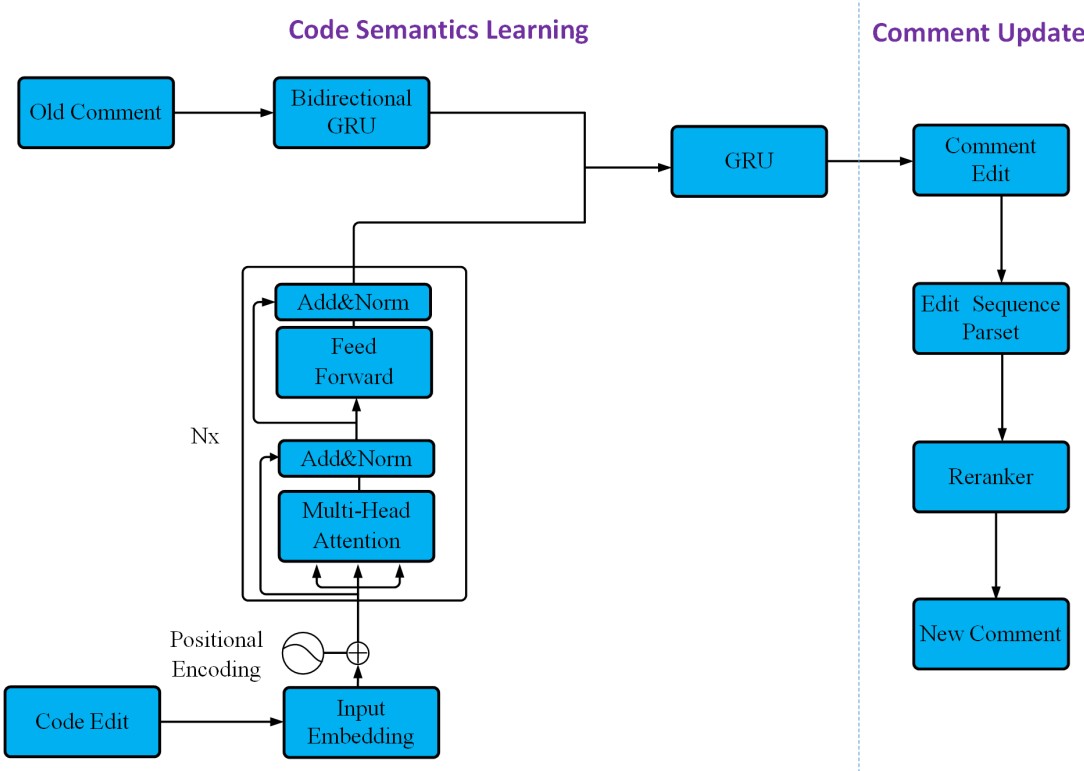

Fig. 1. Overall framework diagram of the code comment update model

comments. As an Encoder, the Transformer takes the code edit $t$ as input and outputs the context vector $h_1$. The bidirectional GRU, serving as another Encoder, takes the old comment $x$ as input and outputs the context vector $h_2$. The two vectors are concatenated to form a vector $H'$, which contains both the code edit and old comment information. This vector is then fed into a GRU for decoding, producing the edited comment $T'$. $T'$ is parsed into an edit sequence and reranking to obtain a new comment $X'$.

### B. Code Semantic Learning Component

The code semantic learning bomponent consists of a couple of crucial components: a Transformer encoder and a bidirectional GRU, working in tandem. The reciprocal GRU is utilized for encoding preexisting comments, whereas the transformer focuses on encoding code modifications. The Transformer integrates a robust self-attentiveness mechanism that adeptly harnesses contextual details within sequences.

To capture the interactions between source code tokens and overcome the obstacles presented by extended and non-temporal dependencies, we employ the position-based encoding mechanism of the transformer and self-attention mechanism. Within each layer of the transformer model, there are two sub-layers, forming its architectural structure: the mechanism of multi-head self-attention and the feedback network with full connectivity.

**Self-attention mechanism:** Here, we focus on describing the self-attention mechanism, which is also the core compo-

nent of CCU. It is also a multi-head attention mechanism [7], and the semantic representation of the basic code for the masked self-attention mechanism does not depend on the absolute position of the mark; in contrast, the interaction between them affects the meaning of the source code. For example, the meanings of the expressions a+b and b+a are identical. The self-attention mechanism abandons a time-series form and focuses on each parameter in the hidden layer. To capture the interdependencies between the input elements, we expand upon the self-attention mechanism to encode the pairwise relationships. For each attention, the source code sequence $x$ is transformed into an output vector sequence $O = \{o_1, \ldots, o_i, \ldots, o_n\}$, where $o_i \in R^{d_k}$, and the expression for $o_i$ is

$$o_i = \sum_{j=1}^{n} \alpha_{ij} \left( x_j \, W^V + a_{ij}^V \right) \tag{1}$$

where $\alpha_{ij} = \frac{\exp e_{ij}}{\sum_{k=1}^{n} \exp e_{ik}}$ and $W^V \in R^{d_{modal} \times d_v}$ are the parameters that are unique to each layer and the attention head, $a_{ij}^V$ is the relative position representation of positions $i$, $e_{ij}$ is computed using a compatibility function that compares two input elements, and the expression for $e_{ij}$ is

$$e_{ij} = \frac{x_i W^Q \left( x_j \, W^K + a_{ij}^K \right)^T}{\sqrt{d_k}} \tag{2}$$

where $a_{ij}^K$ is the relative position representation of position $j$, and the equation is

$$a_{ij}^K = w_{\text{clip}(j-i,k)}^K \tag{3}$$

where $a_{ij}^V$ is the relative position representation of position $i$, and the equation for $a_{ij}^V$ is

$$a_{ij}^V = w_{\text{clip}(j-i,k)}^V \qquad (4)$$

When dealing with linear sequences, edges have the capability to encompass intricate details concerning the positional discrepancies among input elements. The upper limit for the maximum relative position is confined to the absolute value of $k$. The equation for $clip(x,k)$ is

$$clip(x,k) = max(-k, min(k,x)) \qquad (5)$$

The advantage of the self-attention mechanism is that the model can help to examine other positions in the input sequence when processing each word. Unlike RNN, self-attention mechanism only focuses on words with temporal relationships with the current input, thereby solving the problem of the persistent dependence in source code.

**Multi-head attention mechanism:** The multi-head attention mechanism enables the model to concurrently learn multiple self-attention operations, empowering it to highlight diverse facets of information within the source code. The equation for $head_i$ is

$$head_i = Attention(QW_i^Q, KW_i^K, VW_i^V) \qquad (6)$$

where $head_i$ is the intermediate encoding representation of a node in one of the Transformer headers, and the final encoding vector of this node is expressed as $MultiHead(Q,K,V)$. Subsequently, multiple self-attentions are spliced to accurately represent the source code, as shown in Equation 7. The equation for $MultiHead(Q,K,V)$ is

$$MultiHead(Q,K,V) = Concat(head_i, ..., head_h)W^O \qquad (7)$$

where $W^O \in R^{hd_v \times d_{\text{model}}}$.

**Position coding:** The position-coding mechanism uses relative position representation to simulate the paired relationship between source code tags, thereby improving the learning of the source code tags. A sine code is used in the even position. Positional encoding can be formulated as

$$PE(\text{pos}, 2i) = \sin\left(\text{pos}/10000^{2i/d_{\text{model}}}\right) \qquad (8)$$

A cosine code is used in the odd position. The formula is

$$PE(\text{pos}, 2i+1) = \cos\left(\text{pos}/10000^{2i/d_{\text{model}}}\right) \qquad (9)$$

where $pos$ refers to the position of the current word in the sentence and is the index for each value in the pointing quantity.

The Transformer is an encoder, and its input is a code editing sequence. The code editing reflects change to the code, and CCU must capture this change. Each pair of editing operations makes it easier for us to capture code information. Owing to the existence of the editing operation pair, the input Transformer sequence has a back-and-forth connection. For example, in `(keep)public Boolean is unspecified () (keepEnd)`, the `(keep)` and `(keepEnd)` are related to

one another and constitute is a non-continuous word interaction. The word interaction allows the Transformer model to learn the non-timing dependence of code editing, so that the model can better learn source code changes. A code edit is a long sequence, and the self-attention mechanism and position coding used by the Transformer can capture any long-term dependency. The code edit is encoded by a Transformer, and the output is a context vector containing the code editing information.

The bidirectional GRU is an encoder, and its input consists of old comments. After the bidirectional GRU encodes, the output is a context vector that contains the original comment information.

The code semantic learning module comprises a decoder component, alongside other elements. This decoder is a GRU, which consists of an update gate and a reset gate. These gates determine how much previous information should be ignored, and how much new information should be added. The formula for the update gate is

$$z_t = \sigma\left(w_z\left[h_{t-1}; \tilde{x}_t\right] + b_z\right) \qquad (10)$$

where $z_t \in R^m$ represents the update gate, $\sigma$ is the sigmoid function, and $w_z \in R^{m \times (m+n')}$ and $b_z \in R^m$ represent trainable parameters. $\tilde{x}_t \in R^{n'}$ is the adjustment input at time step $t$. The expression for the reset gate is

$$r_t = \sigma\left(w_r\left[h_{t-1}; \tilde{x}_t\right] + b_r\right) \qquad (11)$$

where $r_t \in R^m$, $w_r \in R^{m \times (m+n')}$, $b_r \in R^m$ and the expression for $h_t$ is

$$h_t = z_t \odot h_{t-1} + (1 - z_t) \odot \tilde{h}_t \qquad (12)$$

where $h_t \in R^m$ is the hidden state, $\odot$ is the element-wise multiplication, and the expression for $\tilde{h}_t$ is

$$\tilde{h}_t = \tanh\left(w_h\left[r_t \odot h_{t-1}; \tilde{x}_t\right] + b_h\right) \qquad (13)$$

where $w_h \in R^{m \times (m+n')}$ and $b_h \in R^m$ are trainable parameters.

### C. Comment Update Component

The process of updating comments can be categorized into two primary phases: reevaluating their rankings and parsing the sequence of edits. During parsing, the output of the comment-editing from the decoder is used to perform operations like inserting, deleting, and replacing on the old comments. This process transforms the code edit into a modified comment. Reranking is then performed to generate refined edited comments that preserve the original content and style attributes. To achieve this, two heuristics are employed to reorder candidate sequences during the beam search, and additional priors that are challenging to backpropagate are incorporated.

1) **Generation possibilities:** Since the modification model is exclusively trained on modification operations, it lacks the capability to evaluate the overall coherence and appropriateness of the resultant comments for the modified method. To address this limitation, we incorporated a pre-trained comment

TABLE I
CODE COMMENTS UPDATE DATASET STATISTICS.

| Kinds | Train dataset | Valid dataset | Test dataset |
|---|---|---|---|
| Unique(Code) | 7,271 | 2,473 | 2,690 |
| Mean(Code) | 86.4 | 87.4 | 97.4 |
| Median(Code) | 46 | 49 | 50 |
| Unique(Comment) | 4,823 | 1,695 | 1,737 |
| Mean(Comment) | 10.8 | 11.2 | 11.1 |
| Median(Comment) | 8 | 9 | 9 |
| Examples | 5,791 | 712 | 736 |
| Edit Actions | 8,350 | 1,038 | 1,046 |

generation model into our approach. Trained on a vast corpus of data, this comment generation model is adept at generating novel comments by leveraging provided code snippets. By leveraging this pre-trained model, we can ensure that the generated comments are not only aligned with the code changes but also exhibit high fluency and relevance.

2) **Similarity to old comments:** Until now, the primary focus of CCU's training has revolved around generating accurate modifications for comment updates. However, we also adhere to the principle of minimal modifications in the editing process. To prioritize predictions that efficiently modify comments with minimal alterations, we employed a reranking heuristic based on the similarity to the original comments. Specifically, we utilized the METEOR metric to quantify the resemblance between the candidate parsing predictions and the existing comments. This allows us to prioritize and select candidate edits that closely align with the original comments while achieving the necessary updates.

## III. EXPERIMENT SETUP

### A. Dataset

The dataset used for training and evaluating our model was introduced by Panthaplackel et al. [1] It consists of examples extracted from popular open-source Java projects. The dataset was created by analyzing the submission history of these projects on GitHub. To annotate the old and new code, the javalang library was employed, resulting in a code edit based on the classification of the code segments. As for the comments, they were represented using spaces and punctuation, while HTML tags and "@return" annotations were removed. Code tags were sub-tokenized to handle their presence within comments appropriately. Through these preprocessing steps, the golden editing action sequence for comment editing was derived.

To prevent using examples that are quite similar to each other in training and testing, the items in the training, testing, and validation sets are disjoint, similar to Movshovitz-Attias and Cohen [8]. Of the 7,239 examples in the final dataset, 833 are extracted from the diffs used by Panthaplackel et al. [1]. Table I presents the statistics for the dataset. Including the code and comment tokens that appear at least twice in the training data and the predefined editing keywords, the code and comment vocabulary sizes are 5,945 and 3,642, respectively.

### B. Evaluation Metrics

To assess the quality of generated candidate comments, we employed established performance metrics commonly used in neural machine translation tasks.

**Exact Match.** Exact match pertains to the percentage of cases where the comment generated by the model precisely matches the reference comment, without any disparities. It is a commonly used metric to evaluate tasks related to source code editing. By assessing the exact match rate, we can determine how accurately the model is able to generate comments that align with the desired reference comments, indicating the precision and correctness of the code comment update process [9]. And the equation is

$$xMatch = \begin{cases} 0, if S_{pred} \neq S_{ref} \\ 1, if S_{pred} = S_{ref} \end{cases} \quad (14)$$

Where xMatch represents Exact Match, where $S_{pred}$ represents the generated code comment, and $S_{ref}$ represents the correct code comment.

**BLEU-4.** BLEU [10] is a metric originally used for evaluating the performance of neural machine translation models. It has subsequently gained widespread usage in code-related tasks [11]. BLEU measures the alignment of N-grams between candidate and reference texts. It examines lower-level N-grams to assess word translation accuracy, while higher-level N-grams are utilized to evaluate sentence fluency. BLEU-4 is a variant of the BLEU metric, which calculates the exact match of n-grams of four different lengths. And the equation is

$$BLEU - 4 = BP \cdot \exp(\sum_{n=1}^{4} w_n \log p_n) \quad (15)$$

where, $BP$ represents the penalty factor, $p_n$ represents the n-gram precision, and $w_n$ represents the weights corresponding to the n-gram precision.

**METEOR.** METEOR [12] is a metric that builds upon the foundation of BLEU but introduces several improvements. METEOR employs a weighted harmonic mean and word recall rate to address certain limitations of the BLEU metric. It first determines the optimal alignment between the candidate sentence and the reference sentence, then calculates precision ($P$) and recall ($R$), and finally derives the $F_{mean}$. And the equation is

$$F_{\text{mean}} = \frac{P \cdot R}{\alpha \cdot P + (1 - \alpha)R} \quad (16)$$

where $P$ is the precision, $R$ is the recall, and $\alpha$ is set as a parameter. Then, we calculate the penalty factor, and the calculation of the penalty factor $pen$ is

$$pen = \gamma \left(\frac{c}{m}\right)^{\beta} \quad (17)$$

where $\beta$ and $\gamma$ are the configurable parameters. It can be seen that when there is no phrase match between the candidate sentence and the reference sentence, a single-word match, $c = m$, and the penalty factor is the largest. When the chunk

becomes longer, $c$ decreases and the penalty factor becomes smaller. Therefore, the METEOR is

$$METEOR = (1 - pen) \cdot F_{mean} \quad (18)$$

**SARI.** SARI [13] was initially introduced for assessing text simplification tasks, where it represents the average N-gram F1 scores [14] associated with the delete, add, and retain editing operations. We utilize SARI to gauge the proficiency of our system in learning the editing process. The expression of SARI is

$$SARI = d_1 F_{add} + d_2 F_{keep} + d_3 P_{del} \quad (19)$$

where $d_1 = d_2 = d_3 = \frac{1}{3}$ and the precision of N-gram matches is $P_{\text{operation}}$ . The formula for $P_{\text{operation}}$ is

$$P_{\text{operation}} = \frac{1}{k} \sum_{n=[1,...,k]} p_{\text{operation}(n)} \quad (20)$$

where the expression for recall $R_{\text{operation}}$ is

$$R_{\text{operation}} = \frac{1}{k} \sum_{n=[1,...,k]} R_{\text{operation}(n)} \quad (21)$$

where the expression for $F_{\text{operation}}$ is

$$F_{\text{operation}} = \frac{2 \times P_{\text{operation}} \times R_{\text{operation}}}{R_{\text{operation}} + P_{\text{operation}}} \quad (22)$$

where operation $\in [ \textit{ del, keep, add } ]$, $k$ is the highest N-gram order; this value is set to 4 in our experiments.

### C. Baselines

In this work, we employed three different baselines to compare against our proposed CCU.

(1) **Origin.** Origin directly outputs the old comments without any modifications.

(2) **Unreranking and Bidirectional GRU**. Both the decoder and encoder employ bidirectional GRU. However, the revised comments are not reevaluated and reordered throughout the process.

(3) **Reranking and Bidirectional GRU.** Similar to the second baseline, both the encoder and decoder employ bidirectional GRU.

### D. Training Details

The training process employed the Adam optimizer with an initial learning rate of 0.0001. A batch size of 100 and a dropout rate of 0.6 were used during training. The latent dimensions of the encoder and decoder were configured as 64 and 128, respectively, whereas both comment embedding dimensions and the code were designated as 64. The training objective aimed to minimize the values of negative logarithm. To address the issue of overfitting, we monitored the validation loss, and if there was no decrease observed for ten continuous epochs, we concluded the training process. During the inference stage, a beam search with a width of 20 was utilized to generate potential comments. Our model was implemented in Python 3.6, leveraging the PyTorch framework, and trained using the MindSpore framework. The simulations were conducted on the MindSpore platform, ensuring efficient computation and performance.

TABLE II
DIFFERENT METRICS BLEU, METEOR, SARI, AND EXACT MATCH IN
DIFFERENT BASELINES AND CCU.

| Kinds | BLEU-4 | METEOR | SARI | xMatch |
|---|---|---|---|---|
| Origin | 19.282 | 34.611 | 46.218 | 0.000 |
| No-reranking | 32.109 | 43.359 | 51.16 | 13.723 |
| Reranking | 45.486 | 44.698 | 50.717 | 18.433 |
| CCU | **47.715** | **46.126** | **50.793** | **19.293** |

### IV. EXPERIMENTAL RESULTS

#### A. Comparisons to baselines

**Motivation.** The effect of different natural language processing models in dealing with comment updating may vary due to algorithm, implementation, dataset and other factors. Therefore, it is necessary to design experiments to compare the performance of multiple models in order to find the optimal model to solve the problem.

**Approach.** To assess the effectiveness of CCU, we conducted a comparative analysis with three baselines: Origin (baseline 1), Unreranking and Bidirectional GRU (baseline 2), and Reranking and Bidirectional GRU (baseline 3). To ensure consistency, we retained the original experimental configurations for the baselines, and the chosen evaluation metrics were exact match, BLEU, METEOR, and SARI.

**Results.** We assessed the performance of CCU using four evaluation metrics: exact match, BLEU, METEOR, and SARI. The experimental results are summarized in Table II, which showcases the impact of CCU on these evaluation metrics. Across all metrics, CCU outperformed the other baseline models. For instance, in terms of METEOR, the Origin baseline achieved a score of 34.611, the unreranking baseline achieved a score of 43.359, the reranking baseline achieved a score of 44.698, and CCU achieved a score of 46.126. This represents an improvement of approximately 33.3% over Origin, 6.4% over unreranking, and 3.2% over reranking. The similar results could be achieved in terms of SARI, BLEU-4 and xMatch. These results substantiate the efficacy of CCU in enhancing the quality of revised comments, as it consistently outperformed the baselines, in terms of all four evaluation metrics: BLEU, exact match, METEOR and SARI. This further emphasizes the significance of CCU's capability to effectively model non-temporal dependencies long-term and long-term in code, resulting in enhanced quality of the generated comments.

**Conclusion.** In summary, the results obtained from the three baselines support the argument that CCU is a reasonable and effective approach. The unique attention mechanism integrated into CCU enables it to overcome limitations in capturing complex code relationships, ultimately leading to improved performance across the evaluation metrics.

#### B. Effect of Different Numbers of Layers and Embedding Size

**Motivation.** Parameter settings have a significant impact on the training performance of deep learning models, so it is important to explore the impact of different network parameters (such as Embedding Size, Layer Number, etc.) on model performance.

TABLE III
VARIOUS EVALUATION METRICS INCLUDING BLEU, EXACT MATCH,
METEOR, AND SARI IN DIFFERENT NUMBER OF LAYERS (FIXED
EMBEDDINGSIZE IS 512)

| Number of layers | BLEU-4 | METEOR | SARI | xMatch |
|---|---|---|---|---|
| 2 | 50.641 | 44.460 | 40.323 | 16.984 |
| 3 | **50.793** | **46.126** | **47.715** | **19.293** |
| 4 | 50.285 | 44.782 | 45.819 | 18.750 |
| 6 | 49.555 | 44.174 | 45.981 | 18.342 |

TABLE IV
VARIOUS EVALUATIONS INCLUDING BLEU, EXACT MATCH, METEOR,
AND SARI IN DIFFERENT EMBEDDING SIZE (FIXED LAYER IS 3)

| Embedding size | BLEU-4 | METEOR | SARI | xMatch |
|---|---|---|---|---|
| 64 | 49.653 | 44.639 | 45.939 | 18.214 |
| 128 | 49.889 | 44.624 | 46.212 | 18.342 |
| 256 | 49.995 | 44.715 | 46.527 | 18.342 |
| 512 | **50.793** | **46.126** | **47.715** | **19.293** |

TABLE V
VARIOUS EVALUATION METRICS INCLUDING EXACT MATCH, BLEU,
METEOR, AND SARI IN DIFFERENT CODE SEMANTIC LEARNING
STRUCTURES

| Different structures | METEOR | BLEU-4 | SARI | xMatch |
|---|---|---|---|---|
| Code editing as input to bidirectional GRU | 44.639 | 49.653 | 45.939 | 18.214 |
| Code editing as input to Transformer | **46.126** | **50.793** | **47.715** | **19.293** |

**Approach.** We analyzed the influence of parameters on CCU using the following methods, and we selected the optional hyperparameter values for the model with embedding sizes of 64, 128, 256, and 512, and number of layers of 2, 3, 4, and 6. We used a single variable that remained unchanged, and we adjusted another variable method to determine the influence of the two parameters on the model.

**Results.** It can be observed from the results presented in Tables III and IV that the values of exact match, BLEU, METEOR, and SARI increase as the embedding size increases, under the condition where the control layers remain unchanged. As the embedding size increases, the size of the word embedding becomes larger, so the effect of generating long codes is better, such as longer function names; however, as the embedding size increases, the training time also increases, especially for larger datasets. At times, a larger embedding size may not be optimal. As the size of the layers increases, the performance of the model generally exhibits a downward trend, indicating that more feedforward neural network layers may not necessarily lead to better results, although this increases the complexity of the model. The specific experimental result is that, when the embedding size is 512, the number of layers is 2, 3, 4, and 6, and the best result is obtained when the number of layers is 3. With a fixed layer of 3, when the embedding size is augmented from 64 to 512, there is an improvement in performance across different metrics. The exact match metric shows an increase from 18.214 to 19.293, BLEU rises from 49.653 to 50.793, METEOR sees an increase from 44.639 to 46.126, and SARI improves from 45.939 to 47.715.

**Conclusion.** In summary, the larger the embedding size, the better the performance of the model. In most cases, an increase in the number of layers will cause the performance of the model to degrade slightly.

*C.* Swapping the Order of Transformer and GRU in Code Semantic Learning

**Motivation.** Different semantic learning structures could produced various results, so it is necessary to compare the long-term dependency and non-temporal dependency learning capabilities of Transformers and bidirectional GRUs.

**Approach.** We conducted two sets of experiments. In the first set, the Transformer received old comments as input, while the bidirectional GRU received code editing as input. In the second set, the bidirectional GRU received old comments as input, while the Transformer received code edits as input.

**Results.** We can deduce from Table V that the outcomes derived from inputting code edits into the transformer surpass those obtained from inputting code edits into the dual-directional GRU. The specific results of experiment show that code edits are used as the input for the bidirectional GRU, and code edits are used as the input for the Transformer. A noticeable improvement can be noticed in the accurate match metric, which raise from 18.824 to 19.293. The similar results can be got by the metrics of BLEU, METEOR and SARI.

Long-term dependency problem has always been a common problem in sequence data, especially in the NLP field. LSTM, GRU, and Transformer are commonly used models to solve long term dependency problems. GRU is a highly effective variant of LSTM networks, with a simpler structure and better performance compared to LSTM networks. Therefore it is currently a very manifold neural network model. While GRU shares similarities with LSTM and can address the issue of long-term dependency in recurrent neural networks (RNNs), it is important to note that previous RNN structures, including LSTM and GRU, have partially mitigated the problem of long-term dependence. However, these structures have not completely resolved the challenge of long-term dependency beyond a specific range.

**Conclusion.** The Transformer can better learn the long-term dependency and non-temporal dependency of code edits, for some long sequences or for sequences with close correlations.

## V. RELATED WORKS

This section discusses the consistency check for code and comments, automatic comment generation, and comment quality.

*A. The Consistency Check For Code And Comments*

Code-comment coherence pertains to the extent to which the comment corresponds with the code, ensuring that the comment precisely portrays the functionality of the code. This aspect has garnered significant attention from researchers, particularly in the context of code modifications and maintaining up-to-date comments. The objective is to ensure that

the comments remain relevant and synchronized with the corresponding code changes.

For instance, Jiang et al. [15] conducted a study on function comments in different releases of PostgreSQL spanning from 1996 to mid-2005. Their findings indicated that a decreasing proportion of function comments with titles suggests a lack of timely updates to the documentation interface by developers. In another study, Fluri et al. [16] concentrated on examining the alterations and retentions of source code functions in diverse releases. They employed tree editing operations within the abstract syntax tree (AST) to analyze these changes. They aimed to explore the discrepancies between comments in historical software versions and the source code.

Tan et al. [3] divided comments into two types: comments describing the internal workings of the method in the method body, and comments describing the function specifications of the method in the method header. Sridhara et al. [17] found that the source code itself was correct, and the Javadoc description of the error did not affect the normal execution of the code; however, it would mislead the code user to introduce errors in the use process. To remedy this, Tan et al. [3] proposed a new method for detecting Javadoc comments called tcomment, which dynamically detects comments during software testing. However, this method focuses on the detection of null values and related abnormal method attributes, which are one-sided. Zhou et al. [18] utilized a constraint solver to detect defects in the directives of API documents. In a study conducted by Alghamdi et al. [19], they examined the significance of primitive variable identifiers within comments. A more recent study by Huang et al. [20] introduced a method for automatically identifying outdated source code comments to verify the consistency between the code and its comments.

While existing techniques primarily concentrate on identifying inconsistent or outdated comments, our approach aims to automatically update comments by leveraging code changes. This approach is designed to prevent the introduction of inconsistent and obsolete comments. Therefore, our approach serves as a complementary method to these existing techniques, rather than a direct competitor.

*B. Automatic Comment Generation*

Automated comment-generation technology has emerged as a valuable tool for assisting developers in updating comments. This technology enables the generation of new comments directly from modified code.During the initial phases of research in this domain, the primary emphasis was on approaches based on templates and information retrieval. These approaches involved using heuristic rules to extract relevant information from the code and synthesize comments based on natural language descriptions.

Nevertheless, as deep learning technology has advanced and larger code comment datasets have become available, deep learning-based approaches [21] have emerged as the predominant research direction in addressing this challenge. The application of these methods has resulted in notable enhancements in the precision of automatically generated

comments [22]–[25]. Earlier investigations have put forward diverse approaches based on rules and information retrieval for the generation of comments [26], [27]. As an illustration, Sridhara et al. [17] introduced a method that utilizes summary information embedded in Java source code and predefined templates to generate comments.Another approach, known as ColCom [28], retrieves similar code snippets from open-source projects and utilizes their comments, either reusing them as-is or customizing them, to generate comments for code snippets.

In recent years, there has been a growing trend among researchers to employ probabilistic models for comment generation. For example, Iyer et al. [29] proposed CODE-NN, a neural attention model that generates summaries for C# and SQL code snippets.

LeClair et al. [30] also developed a similar model employing two different encoders to represent code text and SBT sequences for comment generation. Subsequent research has focused on leveraging advances in neural machine translation to address the implicit relationship between code structure information and natural language descriptions, achieving improved comment generation. Yang et al. [31] introduced ComFormer, a novel method based on the Transformer model and fusion hybrid code. Nagata et al. [32] proposed a generation challenge called feedback-comment generation for language learners. Ramin et al. [33] presented API2Com, a model that utilizes application programming interface documentation (API Docs) as a valuable knowledge resource for comment generation. These recent advancements in probabilistic models for comment generation demonstrate a focus on learning the implicit relationship between code structure information and natural language descriptions, leading to notable improvements in comment generation effectiveness.

Prior studies in automatic commit message generation have primarily focused on learning from code changes to produce natural language summaries of those changes [34]. In contrast, our approach centers on applying edits to existing natural language text. Furthermore, we demonstrate that generating comments from scratch is less effective compared to our proposed edit model in the context of comment updates. Kuang et al. [35] introduced a Graph Neural Network-enhanced Transformer model, known as GTrans, to enhance code representation and improve code comprehension. Li et al. [36] proposed an innovative approach based on an enhanced Transformer model for comment generation. This approach effectively addresses long-term dependencies and extracts both textual and structural information from program code.

Compared to previous work, our method enables comments to update with code changes. During the comment update process, in order to better capture the semantic and structural information of the source code, our method uses self attention mechanism and positional encoding mechanism to better improve the quality of comments.By capturing the semantic information of the code, we can understand how the key parts of the code work, and provide more detailed and useful information in the comments. For example, we can capture the input and output of a function, as well as the purpose

and purpose of the function. This helps us better understand the code and manage it better. By capturing the structural information of the code, we can grasp the hierarchical structure and organization of the code. This helps us write better comments and better understand the code.

### C. Comment Quality

Apart from uncovering information encapsulated within the comments, evaluating comments from alternative viewpoints has garnered significant interest among researchers in recent years. For instance, evaluating the quality of comments [37], [38], identifying inconsistencies between code and comments [39], [40], and investigating the co-evolution of comments [41] and code have been areas of focus. The main aim has been to guarantee the consistency between comments and the corresponding code, while upholding a high standard of quality. In recent years, various tools and techniques have been proposed to automatically assess comments based on specific quality attributes and metrics [42]. Nevertheless, there is still a need for a comprehensive model that encompasses the essential quality attributes and metrics for evaluating comments. While previous literature reviews have proposed quality models for software documentation [43]–[45], our focus is specifically on code comments. We aim to develop a unified model that specifically addresses the unique characteristics and requirements of code comments. In assisting developers and researchers to build comment quality assessment tools, Rani [46] provided: (1) a taxonomy for comment convention-related inquiries, which has been empirically validated using data from diverse community forums; (2) a taxonomy of comment information types, also empirically validated, that encompasses various programming languages; (3) a language-agnostic method for automatically detecting these information types; (4) lastly, a comprehensive comment quality taxonomy based on a systematic literature review.

Compared to previous work, our comment quality assessment is more advanced. We adopt a combination of automatic and manual evaluation methods. In the manual evaluation stage, evaluation is conducted from three aspects: fluency, accuracy, and consistency to ensure the scientific nature of comment evaluation.

## VI. CONCLUSION

In this paper, we introduce a novel method named CCU, which addresses the task of automatically updating comments to ensure consistency with code changes. To evaluate the performance of CCU, we conducted comprehensive experiments, employing quantitative analysis as the basis. The experimental results demonstrate that CCU surpasses baseline methods in accurately predicting code transformations.

## DATA AVAILABILITY STATEMENTS

The datasets generated in this work are publicly accessible in the corresponding GitHub repository, and the URL is [47].

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
