# OpenReview forum: "Source Code Changes Just-In-Time Update Via Code Semantics"
_IEEE.org/ICIST/2024/Conference — IEEE ICIST 2024 Conference Submission_

### Official Review · Reviewer_LwSQ · 2024-08-22
**This paper presented a novel approach named Code Comment Update (CCU) model, which incorporates self-attention, positional encoding, and relative positional representation to effectively capture the relationships between different source code tags. The topic of this paper is interesting.**

**Rating:** 10
**Confidence:** 3

**Review:**

Comments to the Author
This paper presented a novel approach named Code Comment Update (CCU) model, which incorporates self-attention, positional encoding, and relative positional representation to effectively capture the relationships between different source code tags. The topic of this paper is interesting. Below is a list of comments that should be taken into account further when revising the paper.
1.	The contribution of this article should be compared with previous literature, and the basic technical difficulties of this article should be listed? And what methods should be used to solve this problem, emphasizing novelty and technological contribution.
2.	In the experimental results section, after separately explaining the three methods, they should be compared uniformly to draw an overall conclusion.
3.	The paper should provide a detailed description of the innovative points to enable readers to quickly understand the article. Meanwhile, please elaborate on the future plans.

---

### Official Review · Reviewer_uvqg · 2024-08-23
**Source Code Changes Just-In-Time Update Via Code Semantics**

**Rating:** 7
**Confidence:** 2

**Review:**

This paper introduced a novel Code Comment Update method, which addressed the task of automatically updating comments to ensure consistency with code changes. The method incorporated self-attention, positional encoding, and relative positional representation to effectively capture the relationships between different source code tags. There are some problems that should be replied. Comments for this submission are given as follows:
1.Please provide a detailed explanation of the role played by the self attention structure in the Code Comment Update model.
2. The paper contains a few minor grammatical errors and formatting inconsistencies. For example, in P3，" The code semantic learning bomponent consists of a couple of crucial components: a Transformer encoder and a bidirectional GRU, working in tandem."
3.In the conclusion section, the limitations of the method proposed in this article should be discussed.

---

### Official Review · Reviewer_CnGj · 2024-08-23
**accept**

**Rating:** 7
**Confidence:** 3

**Review:**

This paper presents a new method called Code Comment Update, which combines self-attention, position encoding, and relative position representation to effectively capture the relationships between different source code labels.The theory is correct and can be accepted after responding the following comments.
(1)In the introduction, it is not enough to state the current work. It should be expended and reconstructed.
(2)There are many typos and grammar errors. The authors should have a native English speaker or software packages to perform the editing check.
(3)The conclusion suggests adding a section on the prospects for future research.

---

### Decision · Program_Chairs · 2024-09-06

Accept (Oral)